# DATA EFFICIENT SUBSET TRAINING WITH DIFFERENTIAL PRIVACY

## ABSTRACT

Private machine learning introduces a trade-off between the privacy budget and training performance. Training convergence is substantially slower and extensive hyper parameter tuning is necessary. Consequently, efficient methods to conduct private training of models have been thoroughly investigated in the literature. To this end, we investigate the strength of the data efficient model training methods in the private training setting. We adapt GLISTER (Killamsetty et al., 2021b) to the private setting and extensively assess its performance. We empirically find that practical choices of privacy budgets are too restrictive for data efficient training to work in the private setting. We make our code publicly available here.

## 1 INTRODUCTION

Machine learning models often memorize training data (Carlini et al., 2023; 2021). In many applications, such as healthcare, finance and generative AI, ensuring privacy of the dataset participants is of utmost importance. Historically, many heuristic methods have been attempted at providing privacy to the dataset participants such as anonymization of the data or removing sensitive columns. These methods have been shown to fail spectacularly in presence of an adversary that can perform *linkage attack* (Dwork et al., 2014) using auxiliary data and reconstruct significant portions of the dataset (Balle et al., 2022). A systematic study in the field of private machine learning was enabled by *differential privacy* due to Dwork et al. (2006).

**Definition 1.1** (($\varepsilon, \delta$)- Differential Privacy)**.** A randomized mechanism $\mathcal{M} : \mathcal{D} \to T$ is ($\varepsilon, \delta$)-differentially private, if $\forall x, x' \in \mathcal{D}$, such that $|x - x'|_1 \le 1$ and $\forall S \subseteq T$, we have that

$$\mathbb{P}\left[\mathcal{M}(x) \in S\right] \le e^{\varepsilon}\left[\mathcal{M}(x') \in S\right] + \delta$$

Where, $|x - x'|_1$ is the $l_1$-norm of the datasets $x, x'$ and the unity bound indicates that they differ in at most one record. $\varepsilon$ and $\delta$ are the privacy loss parameters, higher value indicating lower privacy. By definition, differential privacy ensures that the presence or absence of a single entry in the dataset does not affect output of the mechanism *significantly*. The private analysis in case of machine learning is the computation of gradient with respect to the model weights per sample.

Differential privacy has found large scale adoption in deep learning after the development of the DP-SGD algorithm (Abadi et al., 2016). DP-SGD uses **gradient clipping** and **noising** to induce privacy in training process and a *privacy accountant* tracks the degradation of privacy throughout the training run. With DP-SGD, a model can be trained to achieve decent performance with modest privacy parameters $\varepsilon = 3$ and $\delta \le 1/|D_{train}|$. Though, DP-SGD algorithm poses a significant challenge due to sample gradient clipping which obliterates parallelism by effectively making the batch size equal to 1. Also, large scale problems such as ImageNet classification remain challenging in the private setting (Tang et al., 2024).

In the non-private setting, *data efficient* model training has found much success. It has been shown to maintain the model performance while requiring less data to train. In light of this, we explore the data efficient training paradigm in the private setting. We thoroughly test this paradigm and report our empirical findings here:

- The operations required to extract a high quality training data subset release private information and their privacy budget must be accounted for. Practical privacy budgets $\varepsilon \in [3, 8]$

probe to be extremely restrictive and render the methods for data efficient training impractical.

- We empirically show that the choices of privacy budgets make the search for quality data inefficient and also discuss conditions under which such methods can work.

## 2   RELATED WORK

**Private Machine Learning.** The composition theorems of differential privacy (Dwork et al., 2010) provide components for building more complex mechanisms using simpler ones. The approach taken in most machine learning applications is that of privatizing gradients. DP-SGD (Abadi et al., 2016) first provided a practical implementation of private machine learning, also designing a privacy degradation tracker termed as a *privacy accountant* based on *Rényi divergence* (Mironov, 2017). The work by Gopi et al. (2021) develops a faster algorithm to approximate the bound for k-fold composition of homogeneous DP mechanisms in $O(\sqrt{k})$ time. Kurakin et al. (2022) show that private training performance depends on various factors; larger models are hard to train, hyperparameters tuning is essential and methods like transfer learning boost performance. De et al. (2022) show improvements in performance for training larger models. Moreover, large scale private training such as ImageNet classification remains a challenging task (Tang et al., 2024) with SOTA test accuracy being just 39.39% for $\varepsilon = 8$. Sander et al. (2023) introduce TAN, Total Amount of Noise during training, and use it to inform hyperparameter search for private training. Tang et al. (2024) achieve state of the art performance on multiple datasets across various choices of $\varepsilon$ by phased training with priors learned on noise generated by random processes.

**Data Efficient Training.**   Multiple approaches for data efficient model training have been investigated. One line of work explores iterative subset selection and training approaches and the goal is to find a high quality subset to train (Killamsetty et al., 2021b; Mirzasoleiman et al., 2020; Yang et al., 2023; Killamsetty et al., 2021a). Searching for a high quality subset is a combinatorial problem which is generally solved by optimizing a submodular proxy function. This approach has been used in various domains of machine learning including speech (Wei et al., 2014), vision (Kaushal et al., 2019) and natural language (Ji et al., 2024). Another line of work explores dataset distillation (Chen et al., 2023; Touvron et al., 2021). Yet another line of methods exist exploring dataset pruning by retaining important examples based on their importance scores. Importance score of an example is a function of how often the example is forgotten throughout the course of training (Toneva et al., 2018; Paul et al., 2021). Our work aligns with methods for searching a high quality subset to train models in the private setting.

## 3   PROBLEM FORMULATION AND METHODOLOGY

**Notation.**   Denote the train dataset $\{(x_i, y_i)\}_{i=1}^{|\mathcal{D}|}$ as $\mathcal{D}$ and the validation dataset $\{(x_i, y_i)\}_{i=1}^{|\mathcal{V}|}$ as $\mathcal{V}$. $m_\theta$ denotes a machine learning model parameterized by $\theta \in \mathbb{R}^p$, where $\mathbb{R}^p$ is the parameter space. Let $\ell$ denote an arbitrary loss function. Define the element wise loss function $\ell_i(\theta) := \ell(m_\theta(x_i), y_i)$. Denote the loss on the whole dataset $\mathcal{D}$ as $\mathcal{L}_\mathcal{D}(\theta) := \sum_{i \in \mathcal{D}} \ell_i(\theta)$. We use $\mathcal{M}(\ldots)$ to denote a differentially private mechanism in the following discussion.

### 3.1   PROBLEM FORMULATION

We start by specifying our objective function based on GLISTER by Killamsetty et al. (2021b),

$$\underset{S \subset \mathcal{D}, |S| \leq k}{\arg\min} \ \mathcal{L}_\mathcal{V}(\underset{\theta}{\arg\min} \ \mathcal{L}_S(\theta)) \tag{1}$$

The overall objective consists of two optimization problems. The inner problem optimizes over the model parameters $\theta$, while the outer problem optimizes the val loss over the space of cardinality constrained subsets $S \subseteq \mathcal{D}$ in order to improve model generalization. It is infeasible to solve the above optimization problem directly for general loss functions and we approximate it in the following way. We iterate over the inner and the outer optimization. The inner optimization yields a model $\theta^*(S)$ for a fixed subset $S$. While the outer problem returns the optimal subset $S^*(\theta)$ given

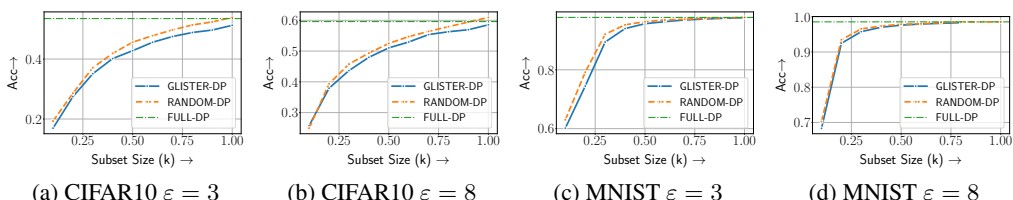

(a) CIFAR10 $\varepsilon = 3$     (b) CIFAR10 $\varepsilon = 8$     (c) MNIST $\varepsilon = 3$     (d) MNIST $\varepsilon = 8$

Figure 1: Performance of GLISTER-DP, RANDOM-DP and FULL-DP on test set for CIFAR10 and MNIST with $\varepsilon \in \{3, 8\}$ across various choices of subset size $k$ as a fraction of $|\mathcal{D}|$

fixed model parameters $\theta$. Solving the inner problem involves gradient descent model training of $m_\theta$ on subset $S$. The outer problem is of combinatorial nature and cannot be solved directly. Killamsetty et al. (2021b) prove that monotone submodular proxy exists for optimizing the outer objective for multiple choice of loss functions and use a greedy algorithm (Mirzasoleiman et al., 2014) to quickly extract a training subset.

### 3.2 DIFFERENTIALLY PRIVATE DATA EFFICIENT TRAINING

As discussed perviouslt, the training procedure iterates over model training and subset selection. We describe how we adapt this non private training method to a private version.

**Differentially Private Training Phase.** We use the DP-SGD algorithm (Abadi et al., 2016) during the training phase. At time step $t$, we use DP-SGD to update model parameters $\theta^t$ by training on the subset $S^t$. The source of privacy leakage during training is through the gradient computation $g(\theta, S) := \nabla_\theta \mathcal{L}_S(\theta)$. DP-SGD performs **gradient clipping** and adds **multidimensional Gaussian noise** to the gradients. The noise scale $\sigma_g$ is based on the privacy parameters $\varepsilon$ and $\delta$ and also depends on the maximum $l_2$ norm of gradients which is bounded to some constant $C$. The privacy accountant tracks the degradation of privacy throughout the training phase. We denote the DP training mechanism $\mathcal{M}_g(\theta, S, g(\cdot)) := g(\theta, S) + p$ where $p \sim \mathcal{N}(0, \sigma_g)$.

**Differentially Private Subset Selection Phase.** The subset selection procedure is reformulated as a submodular maximization problem by Killamsetty et al. (2021b) which can be solved using the stochastic greedy algorithm due to Mirzasoleiman et al. (2014). At its core, an optimal subset $S$ that approximately ($(1 - 1/e)$ approximation guarantee) maximizes a submodular objective function $F$ can be found by greedily choosing an element $e$ with maximum gain $F(S \cup e) - F(S)$ in a sequential manner. We outline the detailed algorithm for differentially private subset selection in Appendix B based on the DP submodular maximization algorithm by Mitrovic et al. (2017), using the exponential mechanism (McSherry & Talwar, 2007) for differential privacy as its core primitive. The argmax step in the greedy algorithm gets replaced by a sampling step based on the exponential mechanism. Overall, the optimization procedure is a k-fold composition of exponential mechanisms, yielding one element at each step. Mitrovic et al. (2017) provide privacy bounds along with approximation guarantees for the overall differentially private submodular optimization algorithm. Denote the DP subset selection mechanism $\mathcal{M}_{ss}(\theta, \mathcal{D}, F(\cdot))$, composed of multiple exponential mechanisms.

**Algorithm.** The detailed description of our training algorithm can be found in Appendix B. We adapt GLISTER (Killamsetty et al., 2021b) by replacing training with DP-SGD and subset selection with the DP submodular maximization algorithm by Mitrovic et al. (2017). We use basic composition for privacy accounting of the two heterogeneous mechanisms $\mathcal{M}_g$ for training and $\mathcal{M}_{ss}$ for subset selection. We refer to our method as GLISTER-DP in our experiments.

**Privacy Accounting.** Privacy accounting during training phase is due to the numerical composition algorithm by Gopi et al. (2021), and runs in $O(\sqrt{k})$ time for k-fold adaptive composition of homogeneous DP mechanisms. The privacy accounting during the subset selection phase is based on the analysis given by Mitrovic et al. (2017). We split the total privacy budget into two parts, $\varepsilon_g$ for training and $\varepsilon_{ss}$ for data subset selection. This follows from the basic composition theorem of DP mechanisms.

## 4 EXPERIMENTS AND DISCUSSION

**Datasets and Baselines.** We experiment with two real world image datasets CIFAR10 and MNIST. We also provide results on class imbalanced synthetic datasets in Appendix C. We compare our GLISTER-DP approach with two baselines. (1) RANDOM-DP selects a training subset $S \subseteq \mathcal{D}$ of size $k$ uniformly at random. RANDOM-DP does not incur any privacy cost during subset selection phase, and the whole budget goes to private training. (2) FULL-DP always trains on the full dataset, and provides a reference for comparison. We test the performance of our methods across various values of $k$, choosing $k \in [0, 1]$ as a fraction of $\mathcal{D}$ and for $\varepsilon \in \{3, 8\}$. Our experiments can be reproduced by running our code.

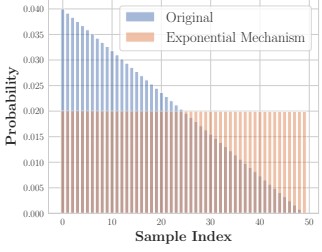

Figure 2: Comparison of original distribution with the one that exponential mechanism samples from. The plot is generated by resampling the true gains and noisy gains and normalizing to produce a valid probability distribution.

**Main Results.** We discuss the main results of our experiments shown in Figure 1. We observe that full training beats both subset selection methods. We also observe that RANDOM-DP outperforms GLISTER-DP for all values of $\varepsilon$ and for both dataset MNIST and CIFAR10. As discussed in Section 3, GLISTER-DP splits the total privacy budget $\varepsilon_{\text{total}}$ into two parts, allocating $\varepsilon_g$ for training and $\varepsilon_{ss}$ for subset selection. Correspondingly, the training noise scale for GLISTER-DP is significantly higher than RANDOM-DP.

During the subset selection phase, GLISTER-DP must make up for the disadvantage of noisier training by choosing a high quality training subset. We show that this is not the case, with help of Figure 2. In the figure, we provide a comparison between the true distribution of the gains of each element and the distribution that the exponential mechanism samples from. The sampling distribution is extremely noisy and it is almost equivalent to sampling elements uniformly at random. Empirically, we observe that the privacy budget $\varepsilon_{ss}$ is too restrictive to yield a good training subset and the generated subset is near random. This explains the loss in performance of GLISTER-DP.

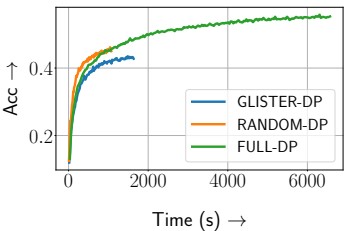

Figure 3: Training convergence for each method on CIFAR10, $\varepsilon = 3$ and $k = 0.5|\mathcal{D}|$.

**Timing Analysis.** In Figure 3, we show the training convergence of each method for $k = 0.5|\mathcal{D}|$ (with $k = |\mathcal{D}|$ for FULL-DP). We observe that RANDOM-DP converges quicker than FULL-DP and GLISTER-DP is the slowest to converge. We observe this trend across all values of $k$ and show this in Appendix D.

**Other Experiments.** In the appendix, we discuss experiments with imbalanced datasets Appendix C. We induce imbalance in real world datasets as well as generate synthetic datasets. We observe that our approach GLISTER-DP performs better than baselines. We also discuss the change in training performance by varying budget allocation between training and subset selection.

## 5 CONCLUSION

In this work, we investigate the potential interaction between data efficient deep learning with differential privacy. To this end, we develop GLISTER-DP, a method for data efficient model training in the private setting based on GLISTER (Killamsetty et al., 2021b). We use DP-SGD (Abadi et al., 2016) for training and differentially private submodular maximization algorithm by Mitrovic et al. (2017) for subset selection. The most essential part of data efficient model training is efficient search of good quality data for training. We empirically observe, that differential privacy poses a significant challenge on the data subset search problem as the privacy budget is too restrictive, rendering it impractical.

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

APPENDIX

## A EXPERIMENTAL DETAILS

The timing numbers are reported on the runs on NVIDIA A6000 GPUs. We do not perform hyperparameter tuning, and run all methods on the same set of hyperparameters in order to reduce computation and expenditure of privacy budget for the same. Throughout our experiments, hyperparameters are chosen so that the noise scale $\sigma$ remains significantly above the "privacy wall" (Sander et al., 2023) and yet allows for model training.

## B GLISTER VS GLISTER-DP

In the following, we compare the original GLISTER algorithm with the DP variant GLISTER-DP. The notable changes in the algorithm are inclusion of the privacy accountants $\text{PA}_{\text{train}}$ and $\text{PA}_{\text{ss}}$ and replacement of the normal training with DP-SGD based private training with $\varepsilon_g$ budget and greedy submodular maximization with the DP version for $\varepsilon_{\text{ss}}$ budget.

---

**Algorithm 1** GLISTER

**Input:** Trainset: $\mathcal{D}$, valset: $\mathcal{V}$, initial subset: $S^0$, initial model: $\theta^0$. LR: $\eta$, epochs: $T$, batch size $B$, selection interval: $L$.
**Output:** Final model $\theta^T$, Final subset $S^T$.

**for** epoch in $1 \ldots T$ **do**
    **if** epoch % L == 0 **then**
        $S^{t+1} \leftarrow \text{GreedyAlgo}(\mathcal{D}, \mathcal{V}, \theta^t, \eta)$
    **else**
        $S^{t+1} \leftarrow S^t$
    **end if**
    $\theta^{t+1} \leftarrow \text{Train}(\theta^t, S^{t+1})$
**end for**
**return** $\theta^T, S^T$

---

**Algorithm 2** GLISTER-DP

**Input:** Trainset: $\mathcal{D}$, valset: $\mathcal{V}$, initial subset: $S^0$, initial model: $\theta^0$. LR: $\eta$, epochs: $T$, batch size $B$, selection interval: $L$, privacy budget $(\varepsilon, \delta)$, allocation ratio $r$.
**Output:** Final model $\theta^T$, Final subset $S^T$.
$\varepsilon_{train} \leftarrow \varepsilon \cdot r$
$\varepsilon_{ss} \leftarrow \varepsilon \cdot (1 - r)$
Initialize $\text{PA}_{\text{train}} \leftarrow \textbf{Accountant}(T, B, \varepsilon_g)$
Initialize $\text{PA}_{\text{ss}} \leftarrow \textbf{Accountant}(T, L, \varepsilon_{ss})$
**for** epoch in $1 \ldots T$ **do**
    **if** epoch % L == 0 **then**
        $S^{t+1} \leftarrow \textbf{DP-GreedyAlgo}(\mathcal{D}, \mathcal{V}, \theta^t, \eta, \text{PA}_{\text{ss}})$
    **else**
        $S^{t+1} \leftarrow S^t$
    **end if**
    $\theta^{t+1} \leftarrow \textbf{DP-Train}(\theta^t, S^{t+1}, \text{PA}_{\text{train}})$
**end for**
**return** $\theta^T, S^T$

---

Due to restricted privacy budget, we perform subset selection every $L$ epochs and use the subset for training for the next $L$ epochs.

## C   CLASS IMBALANCED SYNTHETIC DATASETS

**Real world datasets with induced class imbalance.**   We first present results on class imbalanced real world datasets.  The number of samples for a class vary between 80 percent to 100 percent and is created artificially on the datasets MNIST, CIFAR10 and CIFAR100. We show the results in Table 1. We see that GLISTER-DP outperforms RANDOM-DP on these imbalanced datasets and underlines the utility of the subset selection methods under class imbalanced settings.

| Dataset | Method | $\epsilon$ | $k = 0.1\|\mathcal{D}\|$ | $k = 0.2\|\mathcal{D}\|$ | $k = 0.3\|\mathcal{D}\|$ | $k = 0.4\|\mathcal{D}\|$ | $k = 0.5\|\mathcal{D}\|$ |
|---|---|---|---|---|---|---|---|
| MNIST | RANDOM-DP | 3.0 | 0.6982 | 0.9103 | 0.9474 | 0.9602 | 0.9649 |
| | GLISTER-DP | 3.0 | 0.7231 | 0.9155 | 0.9490 | 0.9609 | 0.9657 |
| | RANDOM-DP | 8.0 | 0.8979 | 0.9599 | 0.9707 | 0.9739 | 0.9760 |
| | GLISTER-DP | 8.0 | 0.9059 | 0.9617 | 0.9710 | 0.9744 | 0.9768 |
| CIFAR-100 | RANDOM-DP | 3.0 | 0.0162 | 0.0249 | 0.0510 | 0.0664 | 0.0854 |
| | GLISTER-DP | 3.0 | 0.0162 | 0.0274 | 0.0483 | 0.0734 | 0.0829 |
| | RANDOM-DP | 8.0 | 0.0344 | 0.0838 | 0.1053 | 0.1337 | 0.1385 |
| | GLISTER-DP | 8.0 | 0.0362 | 0.0810 | 0.1118 | 0.1234 | 0.1433 |
| CIFAR-10 | RANDOM-DP | 3.0 | 0.2872 | 0.3631 | 0.4174 | 0.4460 | 0.4598 |
| | GLISTER-DP | 3.0 | 0.2751 | 0.3719 | 0.4189 | 0.4509 | 0.4669 |
| | RANDOM-DP | 8.0 | 0.3894 | 0.4523 | 0.4856 | 0.5026 | 0.5317 |
| | GLISTER-DP | 8.0 | 0.3878 | 0.4564 | 0.4808 | 0.5111 | 0.5494 |

Table 1: Comparison of performance of RANDOM-DP and GLISTER-DP on mild class imbalance datasets across fraction of training budget

**Experiments with highly imbalanced synthetic dataset.**   Next we provide results on an imbalanced synthetic dataset to illustrate the applicability of data subset selection methods.  We create a synthetic dataset such that it has significant train, val and test distribution shift.  The synthetic dataset contains $N = 5000$ examples, each example having $m = 10$ features and the dataset contains 2 classes. Train dataset has a class imbalance ratio of 1:9, val dataset imbalance ratio is 6:4 and test dataset has an imbalance ratio 9:1. Under these settings, GLISTER-DP has a significant edge over other baselines since the choice of training subset for GLISTER-DP is informed based on the val set as can be seen in Figure 4. As the size of the train subset increases, the performance of both GLISTER-DP and RANDOM-DP become equivalent to FULL-DP.

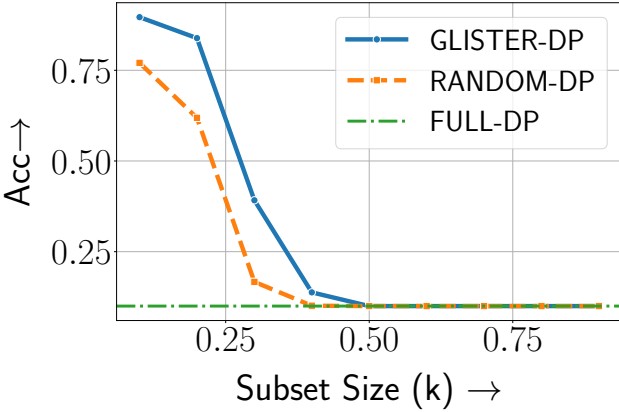

Figure 4: Performance comparison on highly imbalanced synthetic dataset.

# D    TIMING ANALYSIS

We provide the timing analysis of convergence of all the three methods in Figure 5. The following experiment is conducted for CIFAR10 with privacy budget $\varepsilon = 3$ and $\delta = 10^{-5}$. We observe that the training on the random subset give fastest convergence in general. GLISTER-DP converges the slowest across all choices of $k$.

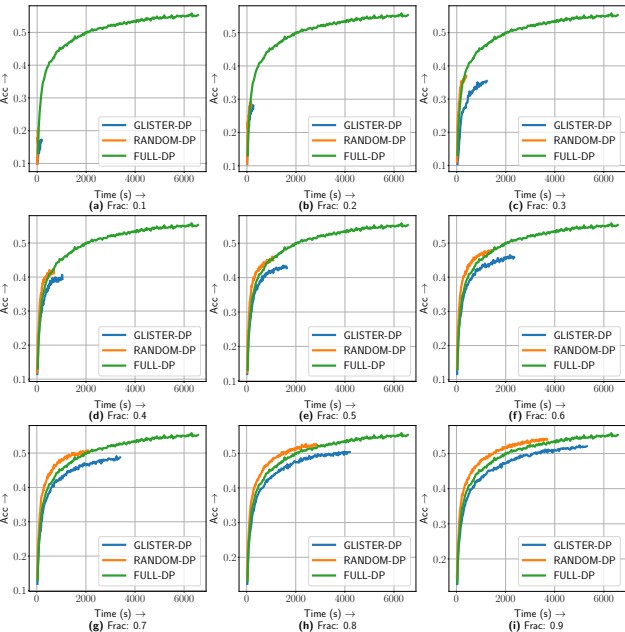

Figure 5: Training convergence plot of GLISTER-DP, FULL-DP and RANDOM-DP across different fractions of training budget on CIFAR-10 $\epsilon = 3$

# E    EXPERIMENTS WITH ALLOCATION RATE.

In Figure 6 we show the effect of budget allocation for GLISTER-DP. Lower allocation rate corresponds to the $\varepsilon_g$ being low, reducing the training privacy budget. We see that the performance of GLISTER-DP monotonically increases as we increase the training budget. Allocating higher budget for subset selection does not improve the subset quality to mitigate the performance degradation during model training. We observe that there is no *sweet spot* in the trade-off between $\varepsilon_g$ and $\varepsilon_{ss}$ and that it is always better to spend privacy budget on training rather than choosing a subset.

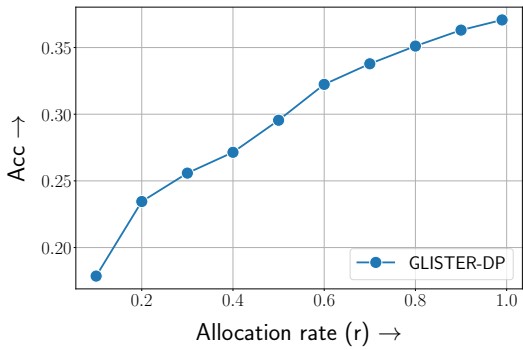

Figure 6: Performance of GLISTER-DP across various choices of budget allocation. $r = 0.1$ corresponds to $\varepsilon_g = 0.1 \cdot \varepsilon_{\text{total}}$

