# OpenReview forum: "Data Efficient Subset Training with Differential Privacy"
_ICLR.cc/2025/Workshop/BuildingTrust — Submitted to BuildingTrust_

### Official Review · Reviewer_RSKH · 2025-02-25
**Data Efficient Subset Training with Differential Privacy**

**Rating:** 5
**Confidence:** 3

**Review:**

**Pros:**
* The paper is well written and clear to me
* The method are efficient and reasonable



**Cons:**
* The baselines are not enough. Only have one full dataset baseline and one subset-based baselines
* The evaluated datasets are too small, only with CIFAR10 and MNIST.
* The performance of the proposed method is not good enough.

---

### Official Review · Reviewer_bdtx · 2025-03-01
**While the paper presents a valuable investigation into the feasibility of data-efficient training in private settings, its empirical results indicate that differential privacy significantly hinders subset selection methods, raising concerns about the practical viability of GLISTER-DP.**

**Rating:** 4
**Confidence:** 4

**Review:**

## Strengths

* The paper addresses balancing differential privacy with data-efficient training, an increasingly relevant subject in privacy-sensitive applications

* The inclusion of RANDOM-DP and FULL-DP as baselines allows for a comparative assessment of the proposed method’s performance under different constraints.

* The authors acknowledge why GLISTER-DP underperforms, discussing the restrictive nature of privacy budgets and the failure of the exponential mechanism in subset selection.

## Weaknesses

* While the paper provides privacy guarantees, it lacks theoretical insights into how GLISTER-DP’s subset selection affects generalization in private settings, relying solely on empirical results.

* The key takeaway is that GLISTER-DP underperforms compared to RANDOM-DP due to excessive noise in subset selection. This diminishes the practicality of the proposed approach.

* While the experiments are detailed, the study focuses primarily on image datasets; it would have been beneficial to see results on text or tabular datasets to generalize conclusions.

* The proposed method is the slowest to converge among the three approaches, further diminishing its practical utility in real-world applications.

---

### Decision · Program_Chairs · 2025-03-04

Reject